# Characteristics of a transgender and gender-diverse patient population in Utah: Use of electronic health records to advance clinical and health equity research

Tiffany F. Ho[1]*, Brian Zenger[2], Bayarmaa Mark[1], Laurel Hiatt[3], Erika Sullivan[1], Benjamin A. Steinberg[4], Ann Lyons[5], Adam M. Spivak[6], Cori Agarwal[7], Marisa Adelman[8], James Hotaling[9], Bernadette Kiraly[1], Sharon Talboys[1]

1 Department of Family and Preventive Medicine, University of Utah School of Medicine, Salt Lake City, Utah, United States of America, 2 Department of Internal Medicine, Washington University in St. Louis, St. Louis, Missouri, United States of America, 3 Department of Human Genetics, Us1niversity of Utah, Salt Lake City, Utah, United States of America, 4 Division of Cardiology, University of Utah School of Medicine, Salt Lake City, Utah, United States of America, 5 Data Science Services, University of Utah, Salt Lake City, Utah, United States of America, 6 Division of Infectious Diseases, University of Utah School of Medicine, Salt Lake City, Utah, United States of America, 7 Department of Surgery, Division of Plastic Surgery, University of Utah, Salt Lake City, Utah, United States of America, 8 Department of Obstetrics & Gynecology, University of Utah, Salt Lake City, Utah, United States of America, 9 Department of Surgery, Division of Urology, University of Utah, Salt Lake City, Utah, United States of America

* tiffany.ho@hsc.utah.edu

**Data Availability Statement:** Data relevant to this study are available from OpenICPSR at https://doi.org/10.3886/E200141V1.

## Abstract

Transgender and gender-diverse (TGD) people, individuals whose gender identity differs from their sex assigned at birth, face unique challenges in accessing gender-affirming care and often experience disparities in a variety of health outcomes. Clinical research on TGD health is limited by a lack of standardization on how to best identify these individuals. The objective of this retrospective cohort analysis was to accurately identify and describe TGD adults and their use of gender-affirming care from 2003–2023 in a healthcare system in Utah, United States. International Classification of Disease (ICD)-9 and 10 codes and surgical procedure codes, along with sexual orientation and gender identity data were used to develop a dataset of 4,587 TGD adults. During this time frame, 2,985 adults received gender-affirming hormone therapy (GAHT) and/or gender-affirming surgery (GAS) within one healthcare system. There was no significant difference in race or ethnicity between TGD adults who received GAHT and/or GAS compared to TGD adults who did not receive such care. TGD adults who received GAHT and/or GAS were more likely to have commercial insurance coverage, and adults from rural communities were underrepresented. Patients seeking estradiol-based GAHT tended to be older than those seeking testosterone-based GAHT. The first GAS occurred in 2013, and uptake of GAS have doubled since 2018. This study provides a methodology to identify and examine TGD patients in other health systems and offers insights into emerging trends and access to gender-affirming care.

**Funding:** (ES and TH) This study was supported by the University of Utah Department of Family and Preventive Medicine, Health Studies Fund, Research Pilot Award. The funders had no role in study design, data collection and analysis, decision to publish, or preparation of the manuscript. (BZ) This study was partially supported by the National Institute of Health (NIH) National Heart, Lung, and Blood Institute (NHLBI) Grant no. 1F30HL149327 https://www.nhlbi.nih.gov/ The funders had no role in study design, data collection and analysis, decision to publish, or preparation of the manuscript.

**Competing interests:** The authors have declared that no competing interests exist.

## Introduction

In recent years, there is increasing recognition of the unique healthcare needs of transgender and gender-diverse (TGD) individuals. In the World Professional Association of Transgender Health (WPATH) Standards of Care for Transgender and Gender Diverse People, Version 8 [1] offers a broad and comprehensive description of people with gender identities or expressions that differ from the gender socially attributed to their sex assigned to them at birth. Estimates of the size of this population vary, but recent studies suggest that approximately 0.6–1% of the general population identifies as TGD [2–4]. As this population seeks healthcare services, it is imperative that research focuses on addressing their specific needs to provide effective, evidence-based care.

Based on small studies with limited datasets, the TGD community experiences more discrimination in the healthcare system than the general population [5,6], and this stigma drives health disparities [5,7]. Transgender and gender-diverse people have higher rates of mental health conditions (anxiety, depression, suicidality, post-traumatic stress disorder) [8–10], substance use disorder [11,12] infectious diseases including HIV/AIDS [13–15] among many other health concerns. These health disparities arise from a complex interplay of factors, including minority stress, stigma, discrimination, barriers to accessing care, and a lack of provider knowledge and cultural competency [7,16–18]. The multifactorial nature of these factors is underscored by the geographic variation in these disparities, which in turn may function as a surrogate for cultural norms (whether affirming or stigmatizing) and healthcare access [19,20]. Many studies on this population rely on small sample sizes [21–23] convenience sampling [2,24–26], or large administrative databases with limited clinical detail [4,27,28] resulting in limited generalizability of findings. Moreover, most of the TGD health research to date has been qualitative or cross-sectional in nature, lacking longitudinal data that could provide insights into the long-term health outcomes and care trajectories of TGD adults.

Given the increased visibility of TGD individuals pursuing transition-related options in the past ten years, it is of critical importance to identify the health needs of TGD individuals particularly in the lens of medical management. The objectives of this study were to characterize the community of TGD adults who received clinical care at a large healthcare system in Utah and to explore the rates of TGD adults who receive gender-affirming hormone-therapy (GAHT) and/or gender-affirming surgeries (GAS). The goal of this manuscript is to share the methodology of creating this cohort so others can apply and improve upon this method to advance understanding of important clinical and health equity questions in the field of transgender health.

## Materials and methods

Before the study started, the authors presented this to the University of Utah Transgender Health Patient and Family Advisory Board (PFAB) with the plan to periodically update the PFAB and obtain feedback and guidance on future projects. The authors included are individuals who provide gender-affirming care; members of the lesbian, gay, bisexual, transgender, queer plus community; and/or allies of the TGD community. The study protocol was reviewed and deemed exempt by the University of Utah's Institutional Review Board (IRB_00159449). Waiver of consent and authorization was approved for this study as this was a retrospective chart review and participants were not contacted. This cohort study followed the Strengthening the Reporting of Observational Studies in Epidemiology (STROBE) reporting guideline [29].

This retrospective cohort study examined clinical encounter data from a Utah-based healthcare system for individuals over the age of 18 years old seeking gender-affirming care. Clinical

**Table 1. ICD-9 and ICD-10 codes used to identify potentially eligible transgender and gender-diverse adults for cohort.**

| ICD-9 Diagnostic Code | ICD-10 Diagnostic Code |
|---|---|
| 301.50 Trans-sexualism with unspecified sexual history (aka 'trans-sexualism not otherwise specified') | F64.0 Transsexualism |
| 302.51 Trans-sexualism with asexual history | F64.1 Dual role transvestism |
| 302.52 Trans-sexualism with homosexual history | F64.2 Gender identity disorder of childhood |
| 302.53 Trans-sexualism with heterosexual history | F64.8 Other gender identity disorders |
| 302.6 Gender identity disorder in children | F64.9 Gender identity disorder unspecified |
| 302.85 Gender identity disorder in adolescents or adults | Z87.890 Personal history of sex reassignment |

and administrative billing diagnosis encounters between 2003 –April 2023 were used to determine the base cohort. Inpatient, outpatient, and procedural visits, as well as medication orders and laboratory results relevant to gender-affirming care were also retrieved. The data was derived from the health system's enterprise data warehouse.

The diagnostic International Classification of Disease (ICD) version 9 and 10 codes specific for "gender dysphoria," and TGD adults were selected based on methodology described in earlier studies [30–33]. Adults aged 18 years and older were included in the dataset if they had at least one clinical encounter that billed specific codes commonly associated with TGD individuals (Table 1).

Surgical procedures were identified based on the Current Procedural Terminology (CPT) codes associated with GAS as listed in Table 2. These codes were collected from both literature review [34,35] and from the surgeons who performed gender-affirming surgeries within one Utah-based healthcare system.

Due to discordance in the type of hormone therapy, gender-affirming surgery and/or gender identity, manual chart review was completed for 101 TGD adults in the GAHT group and 22 were excluded as it was clear their gender identity was concordant with their sex assigned at birth. Thirty-nine adults had prescriptions for both testosterone- and estrogen-based therapy. Of those, five were excluded as they did not meet the criteria for TGD (all postmenopausal cis-women on hormone replacement therapy), four adults had detransitioned to a gender corresponding with their sex assigned at birth, and one adult had active prescriptions for both

**Table 2. Current Procedural Terminology (CPT) codes used for gender-affirming surgeries.**

| Gender-Affirming Surgical Procedures | Top | Bottom |
|---|---|---|
| Feminizing | *Chest Wall Reconstruction (breast augmentation)* 19325 | *Orchiectomy* [35] 54520,54521,54522,54530,54535 *Vulvoplasty* 56805 *Vaginoplasty* 53420, 53430, 54125, 54520, 54690, 55970, 56800, 57291, 57292, 57335 |
| Masculinizing | *Chest Wall Reconstruction (bilateral mastectomies)* 19300,19301,19303,19304, 19318 | *Phalloplasty* 53425, 54660, 55175, 55180, 55980, 57106, 57110 *Hysterectomy +/- oophorectomy* [34] 56307, 56308, 58150, 58152, 58180, 58200, 58210, 58240, 58260, 58285, 58290, 58291, 58541, 58542, 58543, 58544, 58548, 58550, 58552, 58553, 58554, 58570, 58571, 58572, 58573, 58700, 58943, 58950, 58951, 58952, 58953, 58954, 58956 |

injectable testosterone and oral estradiol. To reduce mis-categorization for gender-affirming surgeries, manual chart review was performed for 93 adults who had surgery types that were either incongruent with each other (e.g CPT codes for both vaginoplasty and phalloplasty) or with the category of GAHT (e.g. individual with hysterectomy who was also on estrogen due to endometrial cancer). Thirteen adults from the GAS group were excluded as they did not meet the criteria for TGD. Further review for why such individuals were incorrectly included in this cohort revealed that majority of these individuals had the ICD-10 code Z87.890 (personal history of sex reassignment). Manual chart review was performed for all individuals who were only included in the cohort because they had Z87.890, and no other ICD-9 or –10 diagnosis listed in Table 1. Of the 212 identified, 196 adults were excluded because their gender identity was concordant with the sex assigned at birth.

Manual chart review for a random sample of 30 adults categorized as GAS confirmed with 100% accuracy that the procedures they underwent were indeed gender-affirming. The accuracy of the TGD cohort was verified via a random sample of 50 patients selected based on ICD codes with 100% accuracy. Furthermore, a random sample of 50 known TGD adults were selected from an outpatient clinic list and 100% of the known TGD adults appeared in the ICD based selection.

## Metrics

Transgender and gender-diverse adults were categorized into three groups: those who underwent GAS; those actively managed on GAHT; and those who had the ICD-9 or -10 diagnosis of gender dysphoria but had not undergone active medical therapy or surgical intervention (not actively managed). The specific inclusion criteria for each group are shown in Table 3. Of note, the first two groups are not exclusive of each other. Adults who had any prescription for GAHT were categorized as estrogen-based or testosterone-based hormone therapy (S1 Table). The type of hormone was used to define the broader categories instead of transfeminine and transmasculine to be inclusive of non-binary and gender-diverse adults. Considering that some of these individuals may be receiving active hormone therapy outside of our healthcare system, adults were not included in the actively managed group if they had less than two prescriptions at different dates confirmed within the health system. Thus, adults with two or more unique prescriptions were considered actively managed for GAHT (Fig 1).

The index visit for the cohort was defined as the date of the first prescription of GAHT prescribed in the health care system or date of the first GAS procedure. Of note, the earliest index

**Table 3. Inclusion Criteria for Gender Affirming Hormone Therapy (GAHT) and gender-affirming surgery groups.**

| Study Group | Inclusion Criteria |
|---|---|
| General Testosterone-based GAHT | Adults with at least one prescription for testosterone |
| General Estrogen-based GAHT | Adults with at least one prescription for estrogen |
| Actively Managed Testosterone-based GAHT | Two or more prescriptions of testosterone, each at least one day apart |
| Active Managed Estrogen-based GAHT | Two or more prescriptions of estrogen, each at least one day apart |
| Gender-Affirming Surgery | Any gender affirming surgical interventions as defined in S1 Table |
| Not Actively Managed | Adults with the ICD-9 or ICD-10 code who have not received any medical therapy or surgical intervention. Includes adults who had less than 2 prescriptions of estrogen- or testosterone-based GAHT |

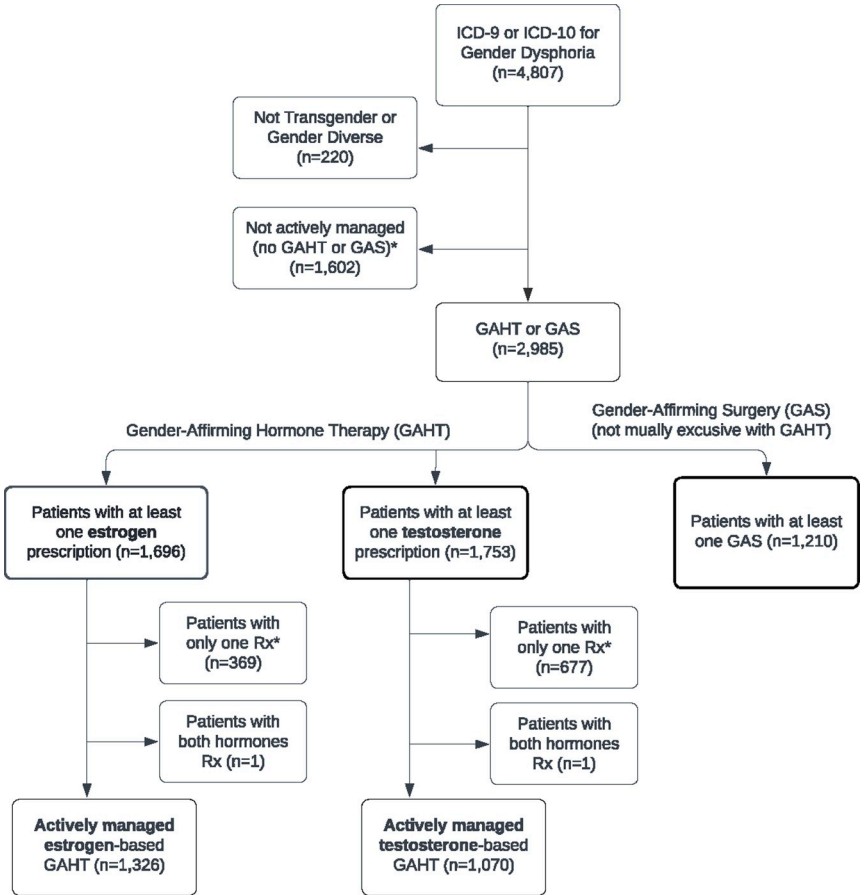

**Fig 1. Flow diagram of categorizing transgender and gender-diverse adults (2003–2023).**

date for GAHT was in 2005 and earliest date for GAS was in 2013. For those who had not undergone medical therapy or surgical treatment, no index date was assigned.

Gender-affirming surgeries were categorized as the following: bottom (i.e. vaginoplasty, orchiectomy, phalloplasty, hysterectomy with or without oophorectomy) or top (i.e. bilateral mastectomy, breast augmentation) as well as feminizing or masculinizing.

Sexual orientation and gender identity (SOGI) data is based on patient self-report, while legal sex (list here as EHR-reported sex) is based on government issued identification. This healthcare system launched the SOGI data questionnaire in 2018 and only recently has had initiatives to standardize and streamline collection of this data. Race and ethnicity were categorized based on the options available in the electronic health record. Body mass index (BMI) was calculated in two ways: the first was a mean derived from the BMI before and closest to the index date and the second was the average of all BMIs prior to the index date. Body mass index was included as a data point as it is one of many metrics used by surgeons to determine eligibility for gender-affirming surgery. Because we were exploring rates of GAS performed, BMI was included. Patients' geographical information was also used and subset into rural, urban, and unknown based on zip code [36].

## Statistical methods

Descriptive statistics are reported as means (standard deviation) or frequencies (percentages). Chi-square and Student t-tests were used to compare the baseline demographics and comorbidities of the cohorts. Fisher's exact test was used when the Chi-square test was inadequate.

Data processing was performed using R (Version 3.6.3), RStudio (Version 1.2.5033), with appropriate packages. Statistical analysis was performed using R (Version 3.4.1), RStudio (Version 1.0.153). A two-sided test with a p-value of less than 0.05 was considered statistically significant. Analysis of the data collected as part of routine clinical care, and subsequent reporting of anonymized, aggregate data, was approved by the University of Utah Institutional Review Board (IRB_00159449).

## Results

Based on the methods specified in Fig 1, the study initially identified 4,807 unique adults who met the ICD-9 or -10 diagnosis of gender dysphoria within this Utah-based health care system from 2003–2023 (Fig 1). After cohort verification with manual chart review, 220 adults were excluded for a final cohort size of 4,587 TGD adults. From 2003–2023, 1,210 TGD adults had undergone at least one GAS, 1,775 had actively received GAHT without undergoing GAS, and 1,602 were not actively managed (had not undergone active medical therapy or surgical intervention).

Table 4 compares the demographics of TGD adults who received GAHT and/or GAS versus those not actively managed. There was no significant difference in race or ethnicity between the two groups. Adults who underwent GAS and/or GAHT were more likely to have commercial insurance (75.0% vs 64.1%) and less likely to have Medicaid (10.4% vs 14.0%), Medicare (3.9% vs 6.9%) and unknown insurance (8.3% vs 12.4%) compared to adults who were not actively managed. Adults who underwent GAS and/or GAHT were more likely to be from Utah (87.2% vs 81.8%) and significantly higher percentage of people from urban areas (92.7% vs 90.6%).

The demographics of TGD adults who received GAHT without GAS was also compared to those who had undergone GAS (Table 5). The mean age at first index date prescription was 26.7 years old for GAHT and 29.7 years old for GAS (p<0.001). There was no significant difference between the two groups in terms of race or ethnicity. The mean BMI was lower for the actively managed GAHT group compared to the GAS group (27.6 vs 28.2, p = 0.051). Adults who received GAHT without GAS were more likely to have commercial insurance (77.0% vs 71.0%) and Medicaid (12.4% vs 7.4%), and less likely to have Medicare (2.9% vs 5.2%) and unknown insurance (4.1% vs 14.4%) compared to the GAS group. Adults in the GAHT without GAS group were more likely to live in-state (95.3% vs 75.4%, p<0.001) and less likely to be from a rural area (6.3% vs 8.9%, p = 0.009) compared to the GAS group.

Table 6 reports the descriptive characteristics for TGD adults actively managed (receiving 2 or more prescriptions) with either estrogen-based (n = 1,326) or testosterone-based (n = 1,070) GAHT. Compared to TGD adults on estrogen-based therapy, individuals on testosterone-based therapy were overall younger (mean age 29.3 vs 25.3 years old, p<0.001), more diverse (Hispanic/Latino: 7.2% vs 13.7%), and more likely to have a higher starting BMI (mean BMI at start of GAHT 26.8 vs 29.0, p<0.001). Adults on testosterone were more likely to have undergone gender-affirming surgery (36.0% vs 17.8%, p<0.001), specifically top surgery (29.3% vs 3.4%) compared to TGD adults on estrogen. For insurance status, those on estrogen-based GAHT were more likely to have Medicaid (12.7% vs 10.6%) and Medicare (4.7% vs 2.1%), and less likely to have commercial insurance (76.2% vs 81.3%) compared to those on testosterone-based GAHT.

**Table 4. Demographics of transgender and gender-diverse adults undergoing gender-affirming surgery and/or gender-affirming hormone therapy versus not actively managed.**

| | Not Actively Managed (N = 1602)[a] | Surgery and/or hormone (N = 2985) | All (N = 4587) | p-value |
|---|---|---|---|---|
| **Race** | | | | 0.193 |
| White | 1224 (86.3%) | 2535 (87.1%) | 3759 (81.9%) | |
| Black or African American | 30 (2.1%) | 46 (1.6%) | 76 (1.7%) | |
| Asian | 20 (1.4%) | 64 (2.2%) | 84 (1.8%) | |
| American Indian/Alaska Native | 30 (2.1%) | 42 (1.4%) | 72 (1.6%) | |
| Native Hawaiian/Pacific Islander | 10 (0.7%) | 19 (0.7%) | 29 (0.6%) | |
| Other | 104 (7.3%) | 206 (7.2%) | 310 (6.8%) | |
| Unknown race | 184 (-) | 73 (-) | 257 (5.6%) | |
| **Ethnicity** | | | | 1.00 |
| Hispanic/Latino | 145 (10.5%) | 299 (10.5%) | 444 (9.7%) | |
| Not Hispanic/Latino | 1235 (89.5%) | 2557 (89.5%) | 3792 (82.7%) | |
| Unknown ethnicity | 222 (-) | 129 (-) | 351 (7.7%) | |
| **Marital status** | | | | < .001 |
| Married/Life partner | 320 (22.7%) | 604 (21.8%) | 924 (20.1%) | |
| Divorced/Legally separated | 68 (4.8%) | 133 (4.8%) | 201 (4.4%) | |
| Widowed | 22 (1.6%) | 11 (0.4%) | 33 (0.7%) | |
| Single | 1001 (70.9%) | 2021 (73.0%) | 3022 (65.9%) | |
| Unknown/Other | 191 (-) | 216 (-) | 407 (8.9%) | |
| **Sex (EHR reported)** | | | | 0.187 |
| Female | 884 (55.5%) | 1570 (52.8%) | 2454 (53.5%) | |
| Male | 701 (44.0%) | 1380 (46.4%) | 2081 (45.4%) | |
| Nonbinary | 9 (0.6%) | 23 (0.8%) | 32 (0.7%) | |
| Unknown sex | 8 (-) | 12 (-) | 20 (0.4%) | |
| **Gender identity** | | | | < .001 |
| Female | 194 (12.1%) | 374 (12.5%) | 568 (12.4%) | |
| Male | 137 (8.6%) | 263 (8.8%) | 400 (8.7%) | |
| Transgender Female | 220 (13.7%) | 756 (25.3%) | 976 (21.3%) | |
| Transgender Male | 261 (16.3%) | 686 (23.0%) | 947 (20.6%) | |
| Non-binary | 201 (12.5%) | 336 (11.3%) | 537 (11.7%) | |
| Unknown gender identity | 589 (36.8%) | 570 (19.1%) | 1159 (25.3%) | |
| **Insurance status** | | | | < .001 |
| Commercial | 1027 (64.1%) | 2239 (75.0%) | 3266 (71.2%) | |
| Medicaid | 224 (14.0%) | 309 (10.4%) | 533 (11.6%) | |
| Medicare | 110 (6.9%) | 115 (3.9%) | 225 (4.9%) | |
| Misc Government | 42 (2.6%) | 75 (2.5%) | 117 (2.6%) | |
| Unknown insurance | 199 (12.4%) | 247 (8.3%) | 446 (9.7%) | |
| **State** | | | | < .001 |
| Idaho | 114 (7.1%) | 113 (3.8%) | 227 (4.9%) | |
| Nevada | 36 (2.2%) | 58 (1.9%) | 94 (2.0%) | |
| Utah | 1309 (81.8%) | 2602 (87.2%) | 3911 (85.3%) | |
| Wyoming | 20 (1.2%) | 41 (1.4%) | 61 (1.3%) | |
| Other states[b] | 122 (7.6%) | 170 (5.7%) | 292 (6.4%) | |
| Unknown state | 1 (-) | 1 (-) | 2 (0.0%) | |
| **Rurality** | | | | 0.017 |
| Urban | 1451 (90.6%) | 2766 (92.7%) | 4217 (91.9%) | |
| Rural | 150 (9.4%) | 218 (7.3%) | 368 (8.0%) | |

*(Continued)*

**Table 4.** (Continued)

|  | Not Actively Managed (N = 1602)[a] | Surgery and/or hormone (N = 2985) | All (N = 4587) | p-value |
|---|---|---|---|---|
| Unknown | 1 (-) | 1 (-) | 2 (0.0%) | |
|  |  |  |  | |

[a]Missing values were not included when calculating the p-values (chi-square test is used to compare categories, t-test is used to compare means).

[b]Other state include: AZ, CA, CO, CT, DC, FL, HI, IA, IL, IN, MA, KY, MD, MI, MN, MO, MS, NC, ND, NE, NH, MT, NM, NY, OH, OK, OR, PA, SC, SD, TN, TX, VA, WA.

Table 7 reports the descriptive characteristics for adults who underwent either feminizing or masculinizing gender-affirming surgery. The mean age of individuals who underwent feminizing surgery was significantly older than those who underwent masculinizing surgery (37.7 years old vs 27.2 years old, p<0.001). More than half (59.0%) of masculinizing surgeries were performed in adults between 21 and 30 years old. Although there was no significant difference in race between surgery type, masculinizing surgery had a higher percentage of adults who identified as Hispanic/Latino (12.3% vs 6.1%, p = 0.006) compared to adults who underwent feminizing surgery. Adults who underwent masculinizing surgery were more likely to have adults with unknown insurance status (18.2% vs 2.4%), more likely to have people from out-of-state (27.1% vs 17.1%), and more likely to have undergone top surgery (90.9% vs 24.9%) compared to those who underwent feminizing surgery.

Table 8 provides a more detailed breakdown regarding the types of GAS adults in this cohort pursued. Masculinizing chest reconstruction surgery (i.e. bilateral mastectomy) was the most commonly performed procedure (n = 875) and phalloplasty was the least commonly performed (n = 46). Fig 2 shows the temporal trends of individuals who underwent gender-affirming surgery from 2013 to 2022 stratified by type of surgery. The number of GAS per year has almost doubled from 2019 to 2022.

## Discussion

This paper outlines the steps utilized to describe adults who identify as TGD via electronic health record data with the goal to use this information as a clinical and equity tool to answer important clinical questions that have largely remained unanswered because previous analyses of other cohorts were insufficiently powered. Almost fifteen percent (14.7%) of adults seeking gender-affirming care lived out of state, and 8.0% lived in rural areas. Almost two-thirds (65.4%) of the total volume of individuals who received GAHT and/or GAS at this healthcare institution had an index date between 2020 –May 2023. The exponential growth in patient volume is reflective of the growing needs of the community as well as the growing number of providers who can provide gender-affirming care within the institution particularly across state lines. The increase in TGD individuals seeking gender-affirming care parallels cultural changes impacting individual disclosure: just as younger individuals are more likely to report being transgender, they are more likely to report feeling comfortable "coming out" and to utilize tools like social media to find community and in turn increase feelings of safety [3,34].

Almost three-quarters (73.5%) of TGD adults had an index date for gender-affirming care (GAHT or GAS) prior to the age of 30. Based on national surveys, researchers estimate the percentage of teenagers and young adults who identify as TGD has doubled in the past five years, with 1.3% of 18- to 24-year-old identifying as TGD [3]. In contrast, 0.5% of individuals 25 to 64 identify as TGD, and 0.3% of individuals 65 and older are TGD [3,22]. This aligns with research findings that a cohort of TGD youths is more likely to disclose their gender identity

**Table 5. Demographics of active gender-affirming hormone therapy (GAHT) and gender-affirming surgery (GAS).**

| | GAHT without GAS[a] (N = 1775) | GAS (N = 1210) | All (N = 2985) | p-value |
|---|---|---|---|---|
| **Age at first index date** | | | | **<0.001** |
| < = 20 | 454 (25.6%) | 174 (14.4%) | 628 (21.0%) | |
| 21–30 | 923 (52.0%) | 645 (53.3%) | 1568 (52.5%) | |
| 31–40 | 245 (13.8%) | 234 (19.3%) | 479 (16.0%) | |
| 41–64 | 142 (8.0%) | 135 (11.2%) | 277 (9.3%) | |
| 65+ | 11 (0.6%) | 22 (1.8%) | 33 (1.1%) | |
| **Age at first index date—Mean (SD)** | 26.7 (9.17) | 29.7 (10.8) | 27.9 (9.99) | **<0.001** |
| **Therapy start year** | | | | **<0.001** |
| 2005–2016 | 139 (7.8%) | 174 (14.4%) | 313 (10.5%) | |
| 2017–2018 | 177 (10.0%) | 201 (16.6%) | 378 (12.7%) | |
| 2019 | 205 (11.5%) | 136 (11.2%) | 341 (11.4%) | |
| 2020 | 270 (15.2%) | 150 (12.4%) | 420 (14.1%) | |
| 2021 | 420 (23.7%) | 208 (17.2%) | 628 (21.0%) | |
| 2022–2023 | 564 (31.8%) | 341 (28.2%) | 905 (30.3%) | |
| **Therapy start year–Mean (SD)** | 2020 (2.48) | 2020 (2.48) | 2020 (2.49) | 1.00 |
| **Race** | | | | 0.354[b] |
| White | 1511 (86.9%) | 1024 (87.2%) | 2535 (84.9%) | |
| Black or African American | 25 (1.4%) | 21 (1.8%) | 46 (1.5%) | |
| Asian | 36 (2.1%) | 28 (2.4%) | 64 (2.1%) | |
| American Indian/Alaska Native | 22 (1.3%) | 20 (1.7%) | 42 (1.4%) | |
| Native Hawaiian/Pacific Islander | 15 (0.9%) | 4 (0.3%) | 19 (0.6%) | |
| Other | 129 (7.4%) | 77 (6.6%) | 206 (6.9%) | |
| Unknown race | 37 (-) | 36 (-) | 73 (2.4%) | |
| **Ethnicity** | | | | 0.684 |
| Hispanic/Latino | 174 (10.2%) | 125 (10.8%) | 299 (10.0%) | |
| Not Hispanic/Latino | 1524 (89.8%) | 1033 (89.2%) | 2557 (85.7%) | |
| Unknown ethnicity | 77 (-) | 52 (-) | 129 (4.3%) | |
| **Body Mass Index (BMI) category[c]** | | | | **<0.001** |
| Underweight (<18.5) | 77 (5.5%) | 26 (2.2%) | 103 (3.5%) | |
| Normal (18.5–24.99) | 560 (39.7%) | 417 (35.1%) | 977 (32.7%) | |
| Overweight (25.0–29.99) | 343 (24.3%) | 345 (29.0%) | 688 (23.0%) | |
| Obese (> = 30) | 429 (30.4%) | 401 (33.7%) | 830 (27.8%) | |
| Unknown BMI | 366 (-) | 21 (-) | 387 (13.0%) | |
| **Body Mass Index—Mean (SD)[c]** | 27.6 (7.89) | 28.2 (6.70) | 27.9 (7.37) | **0.051** |
| **Body Mass Index—Mean (SD)[d]** | 27.4 (7.68) | 28.1 (6.65) | 27.7 (7.23) | **0.019** |
| **Marital status** | | Marital status | | |
| Married/Life partner | 303 (18.5%) | 301 (26.5%) | 604 (20.2%) | |
| Divorced/Legally separated | 76 (4.7%) | 57 (5.0%) | 133 (4.5%) | |
| Widowed | 4 (0.2%) | 7 (0.6%) | 11 (0.4%) | |
| Single | 1251 (76.6%) | 770 (67.8%) | 2021 (67.7%) | |
| Unknown/Other | 141 (-) | 75 (-) | 216 (7.2%) | |
| **Sex (EHR[e] reported)** | | | | **<0.001** |
| Female | 782 (44.2%) | 788 (65.4%) | 1570 (52.6%) | |
| Male | 976 (55.2%) | 404 (33.5%) | 1380 (46.2%) | |
| Nonbinary | 10 (0.6%) | 13 (1.1%) | 23 (0.8%) | |
| Unknown sex | 7 (-) | 5 (-) | 12 (0.4%) | |

*(Continued)*

**Table 5.** (Continued)

| | GAHT without GAS[a] (N = 1775) | GAS (N = 1210) | All (N = 2985) | p-value |
|---|---|---|---|---|
| **Gender identity** | | | | <**0.001** |
| Female | 274 (15.4%) | 100 (8.3%) | 374 (12.5%) | |
| Male | 143 (8.1%) | 120 (9.9%) | 263 (8.8%) | |
| Transgender Female | 579 (32.6%) | 177 (14.6%) | 756 (25.3%) | |
| Transgender Male | 341 (19.2%) | 345 (28.5%) | 686 (23.0%) | |
| Non-binary | 194 (10.9%) | 142 (11.7%) | 336 (11.3%) | |
| Unknown gender identity | 244 (13.7%) | 326 (26.9%) | 570 (19.1%) | |
| **Insurance status** | | | | <**0.001** |
| Commercial | 1380 (77.7%) | 859 (71.0%) | 2239 (75.0%) | |
| Medicaid | 220 (12.4%) | 89 (7.4%) | 309 (10.4%) | |
| Medicare | 52 (2.9%) | 63 (5.2%) | 115 (3.9%) | |
| Misc Government | 50 (2.8%) | 25 (2.1%) | 75 (2.5%) | |
| Unknown insurance | 73 (4.1%) | 174 (14.4%) | 247 (8.3%) | |
| **State** | | | | <**0.001** |
| Idaho | 17 (1.0%) | 96 (7.9%) | 113 (3.8%) | |
| Nevada | 11 (0.6%) | 47 (3.9%) | 58 (1.9%) | |
| Utah | 1691 (95.3%) | 911 (75.4%) | 2602 (87.2%) | |
| Wyoming | 25 (1.4%) | 16 (1.3%) | 41 (1.4%) | |
| Other states[f] | 31 (1.7%) | 139 (11.5%) | 170 (5.7%) | |
| Unknown state | 0 (-) | 1 (-) | 1 (0.0%) | |
| **Rurality** | | | | **0.009** |
| Urban | 1664 (93.7%) | 1102 (91.1%) | 2766 (92.7%) | |
| Rural | 111 (6.3%) | 107 (8.9%) | 218 (7.3%) | |
| Unknown | 0 (-) | 1 (-) | 1 (0.0%) | |
| | | | | |

[a]Missing values were not included when calculating the p-values (chi-square test is used to compare categories, t-test is used to compare means).

[b]Fisher's exact test used.

[c]Mean BMI before and closest to index date.

[d]Mean BMI of all BMI's prior to index date.

[e]EHR = Electronic Health Record.

[f]Other state include: AZ, CA, CO, CT, HI, IL, KY, MD, MO, MS, MT, NM, NY, OH, OK, OR, PA, SC, SD, TN, TX, WA.

in a clinical setting than TGD individuals even slight older: decreased age appears to correspond with increased comfort identifying as TGD. These changes occur in tandem with increased awareness and accessibility of avenues of gender-affirming medical care [37]. Thus, the number of individuals seeking out GAHT is expected to continue to grow. Furthermore, the number of GAS within this healthcare institution has nearly tripled from 2016 to 2019 [38]. Similar to the Wright et al's cohort study, when stratified by age, patients 19–30 years-old had the greatest number of procedures; top surgeries were the most common [38].

Insurance coverage for gender-affirming care, particularly GAS, is variable within the US [39]. In this cohort, 14.4% of all TGD adults who underwent GAS had unknown insurance status, and this increased to 18.2% for those who had masculinizing GAS. These individuals are assumed to be either uninsured or paid out of pocket for GAS services. Compared to the general US population, TGD adults are more likely to be uninsured, unemployed, and living in poverty [40–42]. Even for those with insurance, they commonly face insurance denials for

**Table 6. Demographics based on type of gender-affirming hormone therapy: Estrogen versus testosterone[a].**

| | Estrogen (N = 1326) | Testosterone (N = 1070) | All (N = 2396) | p-value[b] |
|---|---|---|---|---|
| **Age at first hormone prescription** | | | | <0.001[c] |
| 18–20 | 235 (17.7%) | 311 (29.1%) | 546 (22.8%) | |
| 21–30 | 670 (50.6%) | 576 (53.8%) | 1246 (52.0%) | |
| 31–40 | 231 (17.4%) | 137 (12.8%) | 368 (15.4%) | |
| 41–64 | 169 (12.8%) | 44 (4.1%) | 213 (8.9%) | |
| 65+ | 21 (1.6%) | 2 (0.2%) | 23 (1.0%) | |
| **Age at first hormone prescription- Mean (SD)** | 29.3 (11.0) | 25.3 (7.35) | 27.5 (9.78) | <0.001 |
| **Therapy start year** | | | | 0.034 |
| 2005–2016 | 115 (8.7%) | 134 (12.5%) | 249 (10.4%) | |
| 2017–2018 | 176 (13.3%) | 132 (12.3%) | 308 (12.9%) | |
| 2019 | 171 (12.9%) | 131 (12.2%) | 302 (12.6%) | |
| 2020 | 210 (15.8%) | 167 (15.6%) | 377 (15.7%) | |
| 2021 | 315 (23.8%) | 220 (20.6%) | 535 (22.3%) | |
| 2022–2023 | 339 (25.6%) | 286 (26.7%) | 625 (26.1%) | |
| **Therapy start year–Mean (SD)** | 2020 (2.58) | 2020 (2.57) | 2020 (2.57) | 0.222 |
| **Race** | | | | 0.004 |
| White | 1153 (89.3%) | 902 (85.6%) | 2055 (85.8%) | |
| Black or African American | 11 (0.9%) | 25 (2.4%) | 36 (1.5%) | |
| Asian | 25 (1.9%) | 22 (2.1%) | 47 (2.0%) | |
| American Indian/ Alaska Native | 19 (1.5%) | 11 (1.0%) | 30 (1.3%) | |
| Native Hawaiian/Pacific Islander | 12 (0.9%) | 7 (0.7%) | 19 (0.8%) | |
| Other | 71 (5.5%) | 87 (8.3%) | 158 (6.6%) | |
| Unknown race | 35 (-) | 16 (-) | 51 (2.1%) | |
| **Ethnicity** | | | | <0.001 |
| Hispanic/Latino | 91 (7.2%) | 142 (13.7%) | 233 (9.7%) | |
| Not Hispanic/Latino | 1172 (92.8%) | 891 (86.3%) | 2063 (86.1%) | |
| Unknown ethnicity | 63 (-) | 37 (-) | 100 (4.2%) | |
| **Body Mass Index category[d]** | | | | <0.001 |
| Underweight (<18.5) | 53 (5.1%) | 40 (4.5%) | 93 (3.9%) | |
| Normal (18.5–24.99) | 439 (42.5%) | 293 (33.3%) | 732 (30.6%) | |
| Overweight (25.0–29.99) | 275 (26.6%) | 218 (24.8%) | 493 (20.6%) | |
| Obese (> = 30) | 266 (25.8%) | 329 (37.4%) | 595 (24.8%) | |
| Unknown BMI | 293 (-) | 190 (-) | 483 (20.2%) | |
| **Body Mass Index—Mean (SD)[d]** | 26.8 (7.01) | 29.0 (8.31) | 27.8 (7.72) | <0.001 |
| **Body Mass Index—Mean (SD)[e]** | 26.7 (6.91) | 28.7 (8.08) | 27.6 (7.54) | <0.001 |
| Married/Life partner | 250 (20.4%) | 230 (23.0%) | 480 (20.0%) | |
| Divorced/Legally separated | 86 (7.0%) | 33 (3.3%) | 119 (5.0%) | |
| Widowed | 10 (0.8%) | 0 (0%) | 10 (0.4%) | |
| Single | 878 (71.7%) | 736 (73.7%) | 1614 (67.4%) | |
| Unknown/Other | 102 (-) | 71 (-) | 173 (7.2%) | |
| **Sex (EHR[f] reported)** | | | | <0.001[c] |
| Female | 392 (29.7%) | 770 (72.0%) | 1162 (48.5%) | |
| Male | 922 (69.9%) | 290 (27.1%) | 1212 (50.6%) | |
| Nonbinary | 5 (0.4%) | 10 (0.9%) | 15 (0.6%) | |
| Unknown sex | 7 (-) | 0 (-) | 7 (0.3%) | |
| **Gender identity** | | | | <0.001[§] |

(Continued)

**Table 6.** (Continued)

| | Estrogen (N = 1326) | Testosterone (N = 1070) | All (N = 2396) | p-value[b] |
|---|---|---|---|---|
| Female | 335 (25.3%) | 10 (0.9%) | 345 (14.4%) | |
| Male | 17 (1.3%) | 203 (19.0%) | 220 (9.2%) | |
| Transgender Female | 722 (54.4%) | 3 (0.3%) | 725 (30.3%) | |
| Transgender Male | 5 (0.4%) | 534 (49.9%) | 539 (22.5%) | |
| Non-binary | 93 (7.0%) | 161 (15.0%) | 254 (10.6%) | |
| Unknown gender identity | 154 (11.6%) | 159 (14.9%) | 313 (13.1%) | |
| **Insurance status** | | | | **0.002** |
| Commercial | 1010 (76.2%) | 870 (81.3%) | 1880 (78.5%) | |
| Medicaid | 168 (12.7%) | 113 (10.6%) | 281 (11.7%) | |
| Medicare | 62 (4.7%) | 25 (2.3%) | 87 (3.6%) | |
| Misc Government | 39 (2.9%) | 22 (2.1%) | 61 (2.5%) | |
| Unknown insurance | 47 (3.5%) | 40 (3.7%) | 87 (3.6%) | |
| **State** | | | | 0.379 |
| Idaho | 27 (2.0%) | 15 (1.4%) | 42 (1.8%) | |
| Nevada | 12 (0.9%) | 11 (1.0%) | 23 (1.0%) | |
| Utah | 1247 (94.0%) | 998 (93.3%) | 2245 (93.7%) | |
| Wyoming | 15 (1.1%) | 17 (1.6%) | 32 (1.3%) | |
| Other states[g] | 25 (1.9%) | 29 (2.7%) | 54 (2.3%) | |
| **Gender-affirming surgery** | | | | **<0.001** |
| No | 1090 (82.2%) | 685 (64.0%) | 1775 (74.1%) | |
| Yes | 236 (17.8%) | 385 (36.0%) | 621 (25.9%) | |
| **Gender-Affirming surgery type** | | | | **<0.001** |
| Top | 45 (3.4%) | 314 (29.3%) | 359 (15.0%) | |
| Bottom | 134 (10.1%) | 31 (2.9%) | 165 (6.9%) | |
| Both | 57 (4.3%) | 40 (3.7%) | 97 (4.0%) | |
| No surgery | 1090 (82.2%) | 685 (64.0%) | 1775 (74.1%) | |
| **Rurality** | | | | 0.089 |
| Urban | 1253 (94.5%) | 992 (92.7%) | 2245 (93.7%) | |
| Rural | 73 (5.5%) | 78 (7.3%) | 151 (6.3%) | |

[a]Individuals had at least two unique prescriptions for hormone therapy.

[b]Missing values were not included when calculating the p-values (chi-square test is used to compare categories, t-test is used to compare means).

[c]Fisher's exact test used.

[d]Mean BMI before and closest to index date.

[e]Mean BMI of all BMI's prior to index date.

[f]EHR = Electronic Health Record.

[g]Other state include: AZ, CA, CO, CT, HI, IL, KY, MD, MO, MS, MT, NM, NY, OH, OK, OR, PA, SC, SD, TN, TX, WA.

gender-affirming treatments as they are often considered elective cosmetic procedures [39]. Although restrictions for gender-affirming services have been receding, the co-pay can still be cost prohibitive [41]. Further research is needed to explore the differences amongst those with unknown insurance status–who has the finances to pursue gender-affirming care compared to those who cannot access care due to insurance barriers.

Nationally, roughly one in six (16%) TGD individuals live in rural areas [3,43]. This Utah-based cohort data categorized 8.0% of TGD adults living in rural areas, and this percentage increased to 9.4% for adults who met the criteria for TGD but had not sought active GAHT or

**Table 7. Demographics based on type of gender-affirming surgery: Feminizing versus masculinizing.**

| | Feminizing (N = 293) | Masculinizing (N = 917) | All (N = 1210) | p-value[a] |
|---|---|---|---|---|
| **Age at index date[b]** | | | | **<0.001** |
| < = 20 | 11 (3.8%) | 163 (17.8%) | 174 (14.4%) | |
| 21–30 | 104 (35.5%) | 541 (59.0%) | 645 (53.3%) | |
| 31–40 | 86 (29.4%) | 148 (16.1%) | 234 (19.3%) | |
| 41–64 | 72 (24.6%) | 63 (6.9%) | 135 (11.2%) | |
| 65+ | 20 (6.8%) | 2 (0.2%) | 22 (1.8%) | |
| **Age at index date—Mean (SD)** | 37.7 (14.1) | 27.2 (8.05) | 29.7 (10.8) | **<0.001** |
| **Surgery start year** | | | | **<0.001** |
| 2013–2016 | 8 (2.7%) | 166 (18.1%) | 174 (14.4%) | |
| 2017–2018 | 24 (8.2%) | 177 (19.3%) | 201 (16.6%) | |
| 2019 | 37 (12.6%) | 99 (10.8%) | 136 (11.2%) | |
| 2020 | 42 (14.3%) | 108 (11.8%) | 150 (12.4%) | |
| 2021 | 74 (25.3%) | 134 (14.6%) | 208 (17.2%) | |
| 2022–2023 | 108 (36.9%) | 233 (25.4%) | 341 (28.2%) | |
| **Surgery start year–Mean (SD)** | 2020 (1.72) | 2020 (2.59) | 2020 (2.48) | 1.00 |
| **Race** | | | | 0.055[c] |
| White | 255 (89.8%) | 769 (86.4%) | 1024 (84.6%) | |
| Black or African American | 1 (0.4%) | 20 (2.2%) | 21 (1.7%) | |
| Asian | 8 (2.8%) | 20 (2.2%) | 28 (2.3%) | |
| American Indian/Alaska Native | 7 (2.5%) | 13 (1.5%) | 20 (1.7%) | |
| Native Hawaiian/Pacific Islander | 1 (0.4%) | 3 (0.3%) | 4 (0.3%) | |
| Other | 12 (4.1%) | 65 (7.3%) | 77 (6.4%) | |
| Unknown race | 9 (-) | 27 (-) | 36 (3.0%) | |
| **Ethnicity** | | | | **0.006** |
| Hispanic/Latino | 17 (6.1%) | 108 (12.3%) | 125 (10.3%) | |
| Not Hispanic/Latino | 260 (93.9%) | 773 (87.7%) | 1033 (85.4%) | |
| Unknown ethnicity | 16 (-) | 36 (-) | 52 (4.3%) | |
| **Body Mass Index category[d]** | | | | **<0.001** |
| Underweight (<18.5) | 12 (4.1%) | 14 (1.6%) | 26 (2.1%) | |
| Normal (18.5–24.99) | 124 (42.6%) | 293 (32.6%) | 417 (34.5%) | |
| Overweight (25.0–29.99) | 85 (29.2%) | 260 (29.0%) | 345 (28.5%) | |
| Obese (> = 30) | 70 (24.1%) | 331 (36.9%) | 401 (33.1%) | |
| Unknown BMI | 2 (-) | 19 (-) | 21 (1.7%) | |
| **Body Mass Index—Mean (SD)[d]** | 26.6 (6.13) | 28.7 (6.79) | 28.2 (6.70) | **<0.001** |
| **Body Mass Index—Mean (SD)[e]** | 26.6 (6.09) | 28.6 (6.76) | 28.1 (6.65) | **<0.001** |
| **Marriage Status** | | | | |
| Married/Life partner | 85 (29.0%) | 216 (23.6%) | 301 (24.9%) | |
| Divorced/Legally separated | 33 (11.3%) | 24 (2.6%) | 57 (4.7%) | |
| Widowed | 7 (2.4%) | 0 (0%) | 7 (0.6%) | |
| Single | 150 (51.2%) | 620 (67.6%) | 770 (63.6%) | |
| Unknown/Other | 18 (6.1%) | 57 (6.2%) | 75 (6.2%) | |
| **Sex (EHR[f] reported)** | | | | 0.093[c] |
| Female | 191 (65.4%) | 597 (65.4%) | 788 (65.1%) | |
| Male | 101 (34.7%) | 303 (33.2%) | 404 (33.4%) | |
| Nonbinary | 0 (0%) | 13 (1.4%) | 13 (1.1%) | |
| Unknown sex | 1 (-) | 4 (-) | 5 (0.4%) | |

*(Continued)*

**Table 7.** (Continued)

|  | Feminizing (N = 293) | Masculinizing (N = 917) | All (N = 1210) | p-value[a] |
|---|---|---|---|---|
| **Gender identity** |  |  |  | <0.001[c] |
| Female | 86 (29.4%) | 14 (1.5%) | 100 (8.3%) |  |
| Male | 2 (0.7%) | 118 (12.9%) | 120 (9.9%) |  |
| Transgender Female | 174 (59.4%) | 3 (0.3%) | 177 (14.6%) |  |
| Transgender Male | 1 (0.3%) | 344 (37.5%) | 345 (28.5%) |  |
| Non-binary | 9 (3.1%) | 133 (14.5%) | 142 (11.7%) |  |
| Unknown gender identity | 21 (7.2%) | 305 (33.3%) | 326 (26.9%) |  |
| **Insurance status** |  |  |  | <0.001[c] |
| Commercial | 217 (74.1%) | 642 (70.0%) | 859 (71.0%) |  |
| Medicaid | 32 (10.9%) | 57 (6.2%) | 89 (7.4%) |  |
| Medicare | 35 (11.9%) | 28 (3.1%) | 63 (5.2%) |  |
| Misc Government | 2 (0.7%) | 23 (2.5%) | 25 (2.1%) |  |
| Unknown insurance | 7 (2.4%) | 167 (18.2%) | 174 (14.4%) |  |
| **State** |  |  |  | 0.003[§] |
| Idaho | 19 (6.5%) | 77 (8.4%) | 96 (7.9%) |  |
| Nevada | 12 (4.1%) | 35 (3.8%) | 47 (3.9%) |  |
| Utah | 243 (82.9%) | 668 (72.9%) | 911 (75.3%) |  |
| Wyoming | 2 (0.7%) | 14 (1.5%) | 16 (1.3%) |  |
| Other states[g] | 17 (5.8%) | 122 (13.3%) | 139 (11.5%) |  |
| Unknown state | 0 (-) | 1 (-) | 1 (0.1%) |  |
| **Rurality** |  |  |  | 0.893 |
| Urban | 266 (90.8%) | 836 (91.3%) | 1102 (91.1%) |  |
| Rural | 27 (9.2%) | 80 (8.7%) | 107 (8.8%) |  |
| Unknown | 0 (-) | 1 (-) | 1 (0.1%) |  |
| **Surgery type** |  |  |  | <0.001 |
| Top | 73 (24.9%) | 834 (90.9%) | 907 (75.0%) |  |
| Bottom | 156 (53.2%) | 42 (4.6%) | 198 (16.4%) |  |
| Both | 64 (21.8%) | 41 (4.5%) | 105 (8.7%) |  |

[a]Missing values were not included when calculating the p-values (chi-square test is used to compare categories, t-test is used to compare means).

[b]Index date = year of first surgery.

[c] Fisher's exact test used.

[d]Mean BMI before and closest to index date.

[e]Mean BMI of all BMI's prior to index date.

[f]EHR = Electronic Health Record.

[g]Other state include: AZ, CA, CO, CT, HI, IL, KY, MD, MO, MS, MT, NM, NY, OH, OK, OR, PA, SC, SD, TN, TX, WA.

GAS. The discrepancy in the percentage of adults living in rural areas and percentage of those seeking gender-affirming care is attributed to significant health disparities, one of which includes access to healthcare [44–46]. The rapid adoption of telehealth during the COVID-19 pandemic bolstered the total number of TGD visits, for both new and established TGD adults [47]. Increase in access amongst those in rural communities is not as clear. Transgender and gender-diverse adults living in rural areas are twice as likely as their cisgender peers to be uninsured, and rural TGD people of color are three times as likely as their White, cisgender neighbors to be uninsured [45]. Further research is needed to examine how telehealth has affected access to gender-affirming care in rural communities, and how insurance status affects access.

**Table 8. Number of gender-affirming procedures performed between 2013-April 2023.**

| Surgery types | Frequency[a] |
|---|---|
| *Feminizing* | |
| Orchiectomy | 192 |
| Vulvoplasty | 158 |
| Chest Wall Reconstruction (breast Augmentation) | 137 |
| Vaginoplasty | 126 |
| *Masculinizing* | |
| Chest Wall Reconstruction (bilateral mastectomies) | 875 |
| Hysterectomy with/without oophorectomy | 59 |
| Phalloplasty | 46 |
| Other[b] | 186 |

[a]Repeated surgeries are excluded. If a surgery required a revision, it counted as one procedure.

[b]Includes CPT codes for procedures such as vulvectomy, cliteroplasty, scrotoplasty, prostatectomy, urethroplasty.

Research studies focused on answering clinical questions related to gender-affirming hormone therapy have been exponentially growing. Although a national longitudinal cohort study is actively trending the physical, mental and social health data of TGD adults, the results of their study will not be available for years [48]. This delay in evaluating gender-affirming services presents a major barrier for the providers and patients facilitating and receiving care now. Currently, healthcare providers rely on guidelines published by WPATH [1], University of California San Francisco [49], Fenway [50], or the Endocrine Society [51]. These consensus-based guidelines have relied on findings from smaller cohort studies, but the number of larger scale studies is growing. Additionally, studies comparing health outcomes between those who underwent gender-affirming care (surgical or medical therapy) versus not is also growing [38,52,53]. The goal of creating this population cohort is to start answering clinically pertinent questions about patient outcomes as well as identifying health inequities.

A strength and unique aspect of this study is each person was assigned an index date based on their initial gender-affirming hormone therapy prescription or date of first gender-affirming surgery. This will allow future studies to track trends over time including laboratory values, medication dosages, and the incidence and prevalence of certain health conditions. Although, the demographics of this mountain west healthcare system is not a nationally representative sample, it is reflective of the state's demographics. Many studies looking at the population of TGD adults have been concentrated along both coasts, and this represents the first cohort of its kind in a mountain west, conservative state.

## Limitations

Reliance on ICD-9 and ICD-10 codes for gender dysphoria places undue emphasis on medical diagnoses billed by healthcare providers. The number of TGD adults estimated here is likely an underestimate. Some individuals who experience gender dysphoria or gender incongruence may choose not to disclose their concerns to healthcare professionals due to mistrust of the healthcare system [54–56]. Before insurance coverage changes, certain TGD adults explicitly requested that billing codes related to gender dysphoria not be used, as their insurance companies refused to cover these medical services. A few TGD adults who still had parental health insurance also requested different codes because they hadn't disclosed their gender identity yet. Consequently, providers used alternative codes such as endocrine disorder, unspecified. Furthermore, there are individuals who do not identify as TGD but may have been captured in

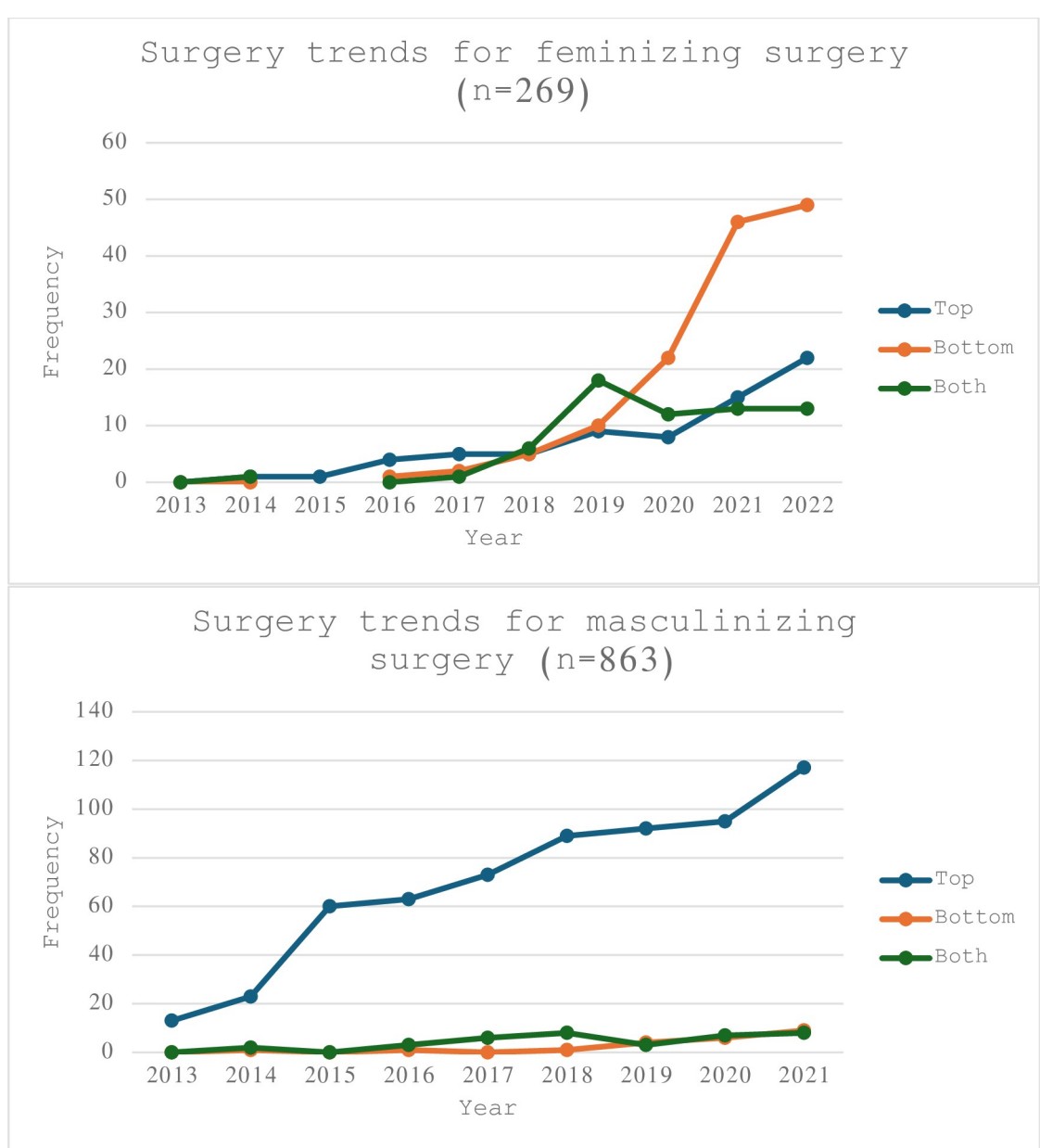

**Fig 2. Gender-affirming surgery trends at a Utah-based healthcare system (2013–2022).**

this cohort due to coding error (such as, but limited to, those captured on manual chart review). While prior studies included adults with ICD-10 Z87.890 (personal history of sex reassignment) [30–33,38], our study found that majority of adults with only this code did not qualify for our TGD cohort. Yet on manual chart review, there was a small handful of individuals who indeed qualify, and we would have missed them if we had excluded this code. Furthermore, the index date assigned is based on the date of the individual's first prescription for estrogen or testosterone within this mountain west healthcare system. This data cannot differentiate if this is the individual's first prescription to start gender-affirming hormone therapy or if the individual had already been on GAHT and was transferring their care to this healthcare system.

## Conclusions

In conclusion, the findings from this dataset underscore a significant alignment with national trends indicating an increase in the number of TGD adults actively pursuing gender-affirming care that necessitates further research to establish best practices across multiple clinical umbrellas. What distinguishes this dataset is the creation of an index date to trend changes both from gender-affirming hormone therapy as well as gender-affirming surgeries. The goal is to utilize this as a clinical and health equity tool further advance the quality of health care provided to the TGD community. There is a critical need for targeted interventions and policy initiatives to bridge the healthcare divide, ensuring equitable access to life-affirming treatments for all, regardless of people's geographic location.

## Supporting information

**S1 Table. List of estrogen- and testosterone- based medications included.**
(DOCX)

## Acknowledgments

Thank you University of Utah Department of Family and Preventive Medicine Research Manager Eliza Taylor, MPH for the formatting of this manuscript.

## Author Contributions

**Conceptualization:** Tiffany F. Ho, Brian Zenger, Erika Sullivan, Benjamin A. Steinberg, Adam M. Spivak, Sharon Talboys.

**Data curation:** Tiffany F. Ho, Brian Zenger, Bayarmaa Mark, Benjamin A. Steinberg, Ann Lyons, Sharon Talboys.

**Formal analysis:** Bayarmaa Mark, Ann Lyons, Sharon Talboys.

**Funding acquisition:** Tiffany F. Ho, Brian Zenger, Erika Sullivan, Benjamin A. Steinberg.

**Investigation:** Tiffany F. Ho, Brian Zenger, Bayarmaa Mark, Erika Sullivan, Benjamin A. Steinberg, Adam M. Spivak, Sharon Talboys.

**Methodology:** Tiffany F. Ho, Brian Zenger, Bayarmaa Mark, Sharon Talboys.

**Project administration:** Tiffany F. Ho, Brian Zenger, Laurel Hiatt, Sharon Talboys.

**Resources:** Tiffany F. Ho, Erika Sullivan, Ann Lyons, Adam M. Spivak, Cori Agarwal, Sharon Talboys.

**Software:** Bayarmaa Mark, Ann Lyons.

**Supervision:** Tiffany F. Ho, Brian Zenger, Erika Sullivan, Sharon Talboys.

**Validation:** Tiffany F. Ho, Bayarmaa Mark.

**Visualization:** Tiffany F. Ho, Bayarmaa Mark, Sharon Talboys.

**Writing – original draft:** Tiffany F. Ho, Brian Zenger, Sharon Talboys.

**Writing – review & editing:** Tiffany F. Ho, Brian Zenger, Bayarmaa Mark, Laurel Hiatt, Erika Sullivan, Benjamin A. Steinberg, Ann Lyons, Adam M. Spivak, Cori Agarwal, Marisa Adelman, James Hotaling, Bernadette Kiraly, Sharon Talboys.

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
