## [Decision Letter · Decision Letter 0]

13 Feb 2024

PONE-D-23-40640Characteristics of a transgender and gender-diverse patient population in Utah: Use of electronic health records to advance clinical and health equity researchPLOS ONE

Dear Dr. HO,

Thank you for submitting your manuscript to PLOS ONE. After careful consideration, we feel that it has merit but does not fully meet PLOS ONE’s publication criteria as it currently stands. Therefore, we invite you to submit a revised version of the manuscript that addresses the points raised during the review process.

We look forward to receiving your revised manuscript.

Kind regards,

Daniel Antwi-Amoabeng, MD, MSc

Academic Editor

PLOS ONE

Journal Requirements:

Additional Editor Comments:

The authors have tackled an important subject matter, and the work is worthy of publication in PLOS ONE once the reviewers’ comments are addressed.

Reviewers' comments:

Reviewer's Responses to Questions

**Comments to the Author**

1. Is the manuscript technically sound, and do the data support the conclusions?

Reviewer #1: Yes

Reviewer #2: Yes

2. Has the statistical analysis been performed appropriately and rigorously? 

Reviewer #1: Yes

Reviewer #2: Yes

3. Have the authors made all data underlying the findings in their manuscript fully available?

Reviewer #1: Yes

Reviewer #2: Yes

4. Is the manuscript presented in an intelligible fashion and written in standard English?

Reviewer #1: Yes

Reviewer #2: Yes

5. Review Comments to the Author

Reviewer #1: I thank the authors for establishing this important cohort and describing in the associated manuscript.

In general it is well written and I think it fits with the journals overall scope, and in particular a benefit to society.

I have some specific comments which can be found in the attached annotated manuscript.

However, two broader comments to be addressed by the authors are:

1. Please can the authors describe in the methods to what degree have TGD community been involved in the design and conduct of the research, and how will they be involved in future cohort analysis and dissemination of results.

2. The different gender affirming surgeries have quite different rates of uptake which also vary within healthcare systems. This granularity is seldom published and would be valuable not only to scientists and physicians, but to policy makers. For this reason, describing the frequency of GAS in smaller categories e.g. hysterectomy BSO vs phallo/metoidio vs chest would be helpful.

I would recommend that the authors resubmit having addressed these comments.

Reviewer #2: I would like to thank the authors for the opportunity to review their manuscript. This study provides a blueprint for how to identify and describe TGD adults and their usage of gender-affirming care within a health care system based on ICD codes. While I am recommending revisions, my hope is that this article reaches publication given its importance.

Overall

I am curious about the decision to include BMI. I understand that many surgeons have BMI requirements for accessing surgeries, but it seemed like an add-on. There was no mention of BMI/weight in the lit review or the discussion, making me question its importance to your study. Research has shown the BMI to be an outdated measure, and further research has shown that the use of the BMI can be used to (1) gatekeep surgeries, and (2) contribute to weight stigma, particularly for folks in larger/fat bodies.

Abstract

Delete the word “have” in this sentence: “During this time frame, 2,995 adults have received gender-affirming hormone therapy (GAHT)…”

Introduction

Line 80: I recommend choosing a different word than “TGD experience”

The introduction/background is informative and grounds the reader for what’s coming next.

Methods

Line 117: “…and 22 were excluded as they did not meet the criteria for TGD.” This sentence confused me because it left me wondering what the criteria is for TGD. In line 119, you write something similar, except in parenthesis you include, “(all postmenopausal cis women on hormone replacement therapy).” Will you clarify prior to line 117 what the criteria are for TGD in this study? And if criteria aren’t met, will you specify why that is like you did in line 119? I’m also wondering if this could be an issue with phrasing (“criteria for TGD”).

Lines 112-125: I recommend re-writing this paragraph in a way that is more accessible to the reader. I found myself a bit lost reading this paragraph, particularly around the cohort being TGD-identified with 100% accuracy and then some folks not meeting criteria for TGD. In order for these methods to be replicated, I think there needs to be a bit more clarity.

I appreciate the way your team used the specific hormone therapy as categories to be more inclusive of nonbinary and TGD folks.

The statistical methods section makes sense and flows well.

Results

In your tables, I recommend removing the word Caucasian since it’s outdated.

The tables look great and really support the results section.

Discussion

Line 267: delete “an”

In the discussion, I think it could be worthwhile to note that although there are more known trans people, trans people have always existed and may not have felt comfortable/safe being as visible as they are today. Access to care is becoming more widespread and known, making it more likely for trans folks to engage with the health care system to access gender-affirming care. I think it’s important for readers to understand that there isn’t necessarily an increase in trans people, but there is an increase in trans visibility and trans people accessing care.

Conclusion

Lines 344-346: “In conclusion, the findings from this dataset underscore a significant alignment with national trends indicating a rapid increase in the number of TGD adults actively pursuing gender-affirming care.” I recommend re-writing this sentence. Unintentionally, by using language such as “rapid increase,” this may fuel arguments around social contagion and push the narrative that the number of TGD people is increasing rapidly, and the number of folks seeking gender-affirming care is rapidly increasing. It is clear that this is not your intention, but I immediately had this thought when reading this.

The closing sentence of the conclusion is great!

Again, thank you for the opportunity to review your manuscript. I can tell how intentional this work was conducted, and the thoughtfulness of the research team. I am recommending revisions to increase clarity and accessibility for readers.

6. PLOS authors have the option to publish the peer review history of their article (what does this mean?). If published, this will include your full peer review and any attached files.

Reviewer #1: **Yes: **Dr Alison Berner

Reviewer #2: **Yes: **Brendon Holloway

---

## [Author Response · Author response to Decision Letter 0]

10 Apr 2024

Thank you for reviewing our manuscript titled “Characteristics of a transgender and gender-diverse patient population in Utah: Use of electronic health records to advance clinical and health equity research” (Manuscript ID PONE-D-23-40640). We appreciate your comments and the reviewer’s constructive feedback. We have addressed all comments and revised the manuscript accordingly. Enclosed is a point-by-point response (submitted as "Response to Reviewers"

---

## [Editor Report · Decision Letter 1]

16 Apr 2024

Characteristics of a transgender and gender-diverse patient population in Utah: Use of electronic health records to advance clinical and health equity research

PONE-D-23-40640R1

Dear Dr. HO,

We’re pleased to inform you that your manuscript has been judged scientifically suitable for publication and will be formally accepted for publication once it meets all outstanding technical requirements.

Kind regards,

Daniel Antwi-Amoabeng, MD, Msc

Academic Editor

PLOS ONE

Additional Editor Comments (optional):

Thank you for addressing the reviewers' comments. The work is accepted for publication in the revised form.
---

## [Editor Report · Acceptance letter]

26 Apr 2024

PONE-D-23-40640R1 

PLOS ONE

Dear Dr. Ho, 

I'm pleased to inform you that your manuscript has been deemed suitable for publication in PLOS ONE. Congratulations! Your manuscript is now being handed over to our production team.

Kind regards, 

on behalf of

Dr. Daniel Antwi-Amoabeng 

Academic Editor

PLOS ONE